# *Rhodococcus* Strains from the Specialized Collection of Alkanotrophs for Biodegradation of Aromatic Compounds

**DOI:** 10.3390/molecules28052393

**Published:** 2023-03-05

**Authors:** Anastasiia Krivoruchko, Maria Kuyukina, Tatyana Peshkur, Colin J. Cunningham, Irina Ivshina

**Affiliations:** 1Perm Federal Research Center, 13a Lenin Street, 614990 Perm, Russia; 2Microbiology and Immunology Department, Perm State University, 15 Bukirev Street, 614068 Perm, Russia; 3Department of Civil and Environmental Engineering, University of Strathclyde, James Weir Building, Level 5, 75 Montrose Street, Glasgow G11XJ, UK

**Keywords:** *Rhodococcus* bacteria, biodegradation, mono- and polycyclic aromatic hydrocarbons, phenol, pyridine, functional genes

## Abstract

The ability to degrade aromatic hydrocarbons, including (i) benzene, toluene, *o-*xylene, naphthalene, anthracene, phenanthrene, benzo[a]anthracene, and benzo[a]pyrene; (ii) polar substituted derivatives of benzene, including phenol and aniline; (iii) N-heterocyclic compounds, including pyridine; 2-, 3-, and 4-picolines; 2- and 6-lutidine; 2- and 4-hydroxypyridines; (iv) derivatives of aromatic acids, including coumarin, of 133 *Rhodococcus* strains from the Regional Specialized Collection of Alkanotrophic Microorganisms was demonstrated. The minimal inhibitory concentrations of these aromatic compounds for *Rhodococcus* varied in a wide range from 0.2 up to 50.0 mM. *o-*Xylene and polycyclic aromatic hydrocarbons (PAHs) were the less-toxic and preferred aromatic growth substrates. *Rhodococcus* bacteria introduced into the PAH-contaminated model soil resulted in a 43% removal of PAHs at an initial concentration 1 g/kg within 213 days, which was three times higher than that in the control soil. As a result of the analysis of biodegradation genes, metabolic pathways for aromatic hydrocarbons, phenol, and nitrogen-containing aromatic compounds in *Rhodococcus*, proceeding through the formation of catechol as a key metabolite with its following ortho-cleavage or via the hydrogenation of aromatic rings, were verified.

## 1. Introduction

Aromatic compounds, such as mixtures of benzene, toluene, ethylbenzene, and xylene isomers (BTEX), polycyclic aromatic hydrocarbons (PAHs), phenol, aniline, phthalates, pyridine, quinoline, coumarin, and their derivatives, represent a large group of dangerous ecopollutants. These compounds are released into the environment as a result of oil spills; leaks and wastes from petrochemical and synthetic industries including the production of fuels, refinery products, explosives, resins, plasticizers, pesticides, herbicides, insecticides, paints, pharmaceuticals, dyes, cosmetics, textile processing, and leather manufacturing; leaching from landfills; releases via tail gas from automobile exhausts; transport leaks; combustion; anthropogenic activities at gas stations, steel-making factories, and the coal chemical industry [1,2,3,4,5,6,7,8,9,10,11,12]. Aromatic pollutants exhibit carcinogenic, genotoxic, cytotoxic, phytotoxic, teratogenic, mutagenic, DNA-damaging, and enzyme-inhibiting effects; trigger cell aggregation; cause bone marrow diseases; induce reproductive disorders; act as solvents for cell components or as endocrine-disrupting agents, and vapors can cause eye, nose, and throat irritation as well as headaches, dizziness, and nausea [4,5,7,8,9,12,13]. The development of efficient and available methods for the complete degradation of these pollutants and remediation of biotopes contaminated with arenes is required [10].

Bioremediation, which exploits the metabolic properties of pollutant-degrading microorganisms for the elimination of toxic compounds in contaminated sites, is a well-known ecologically safe and cost-efficient approach [14,15,16,17]. To metabolize aromatic pollutants, microorganisms use a complex of adaptations including mechanisms of resistance against chemical toxicants, systems for the cellular transport of pollutants, and specific catalytic enzymes. On the one hand, arenes are recalcitrant molecules as they have aromatic rings, which are chemically stable conjugated π–electron systems. On the other hand, the metabolism of all arenes is unified, resulting in the formation of universal metabolites such as catechol, protocatechuate, gentisate, homogentisate, or benzaldehyde with further ortho- or meta-cleavage of aromatic rings [17,18,19,20]. Non-specific enzymes determine the ability of bacteria to simultaneously degrade various aromatic compounds, for example, PAHs and BTEX, or PAH mixtures [19,21,22]. At the same time, the specific enzymes are responsible for the initial transformation of aromatic molecules.

Various microorganisms can degrade aromatic compounds. Among them are bacteria (members of *Acinetobacter*, *Achromobacter*, *Aeromonas*, *Agromyces, Alcaligenes*, *Arthrobacter*, *Bacillus*, *Burkholderia*, *Comamonas*, *Delftia*, *Halomonas*, *Lactobacillus*, *Nocardiodes*, *Ochrobactrum*, *Pseudomonas, Pseudonocardia*, *Rhodococcus,* etc.) and fungi (members of *Aspergillus*, *Cunninghamella*, *Glomerella*, *Saccharomyces*, etc.). The biodegradation abilities of microbial communities and pure cultures toward toxic arenes are described [1,3,6,11,12,20,23,24,25,26]. A promising group of aromatic biodegraders is the extremotolerant bacteria of the genus *Rhodococcus* Zopf 1891 (Approved Lists 1980; domain ’*Bacteria*’, phylum *Actinomycetota*, class *Actinomycetes*, order *Mycobacteriales*, family *Nocardiaceae*; https://lpsn.dsmz.de/genus/rhodococcus, accessed on 2 March 2023). *Rhodococcus* spp. are isolated from many polluted areas; dominate in hydrocarbon-contaminated ecosystems; are able to metabolize a wide range of emerging pollutants including petroleum hydrocarbons, polychlorinated biphenyls, pharma pollutants, pesticides, explosives, flame retardants, plasticizers, defoliants, dyes, and microplastics; are resistant to multiple stresses; are able to maintain high metabolic activities under adverse conditions [15,16,27,28,29,30]. *Rhodococcus* bacteria can degrade BTEX, PAHs, phenol, phthalates, phthalic esters, pyridine, quinoline, and complex aromatic compounds [6,24,29,30,31,32,33,34,35]. Most studies describe individual *Rhodococcus* strains isolated from environmental samples or, more rarely, their associations. However, a comparative investigation of the ability of various *Rhodococcus* species to degrade diverse aromatic compounds has not yet been performed.

In ecobiotechnology, the biological and ecological safeties of bacterial cultures are important parameters. The use of strains not identified at the species level (*Rhodococcus* sp.) for bioremediation causes risks of pathogenic or phytopathogenic strains entering the environment. To minimize these risks, it is recommended to order strains from well-known microbial collections [15]. Microbial culture collections registered with the World Data Centre for Microorganisms (WDCM, http://wdcm.nig.ac.jp, accessed on 2 March 2023) are powerful sources of properly characterized and biotechnologically promising strains with high metabolic activities. Cultures deposited there are well described in terms of taxonomy and pathogenic/phytopathogenic properties with relevant information on strain features (isolation and growth conditions, utilized substrates, adhesive activities, tolerance, etc.) [36]. One such collection is the Regional Specialized Collection of Alkanotrophic Microorganisms (acronym IEGM, www.iegmcol.ru, accessed on 2 March 2023, WFCC 285), where the largest fund of pure, identified, and well-described hydrocarbon-oxidizing isolates belonging to the genus *Rhodococcus* is maintained [37].

The aim of this study was to estimate the metabolic capabilities of *Rhodococcus* strains from the IEGM Collection to degrade a variety of aromatic compounds.

## 2. Results

### 2.1. Toxicity of Aromatic Compounds for Rhodococcus

The toxicities of aromatic compounds were estimated using the reference strain *R. ruber* IEGM 231, which was previously studied as a degrader of a wide range of hydrocarbons and as a producer of biotechnologically important metabolites [38]. As shown in Table 1, the minimal inhibitory concentrations (MICs) of used aromatic compounds broadly varied between 0.2 and 50.0 mM. Two groups of aromatic substances were clearly distinguished: relatively low toxicity (MIC = 25.0–50.0 mM) and high toxicity (MIC = 0.2–0.8 mM). Most of the tested compounds (*o-*xylene, all PAHs, *o-*phthalic acid, salicylic acid, phenol, aniline, pyridine, methyl- and hydroxy-substituted derivatives of pyridine, and coumarin) constituted the first group. The second group contained benzene, toluene, and meta- and para-isomers of phthalic acid. The toxicities of aromatic compounds from the second group were 31–250 times higher than those from the first group. Moderate toxicity (MIC = 6.3 mM) was revealed for only one compound, a two-ring N-heterocycle quinoline (Table 1).

### 2.2. Degradation of Monoaromatic Hydrocarbons by Rhodococcus

Strains of the most represented species (*R. erythropolis/R. qingshengii*, *R. fascians*, *R. opacus*, and *R. ruber*) were screened for their abilities to use benzene and its less toxic homolog with two methyl groups in the ortho*-*position, *o-*xylene. The growth of *Rhodococcus* cells in the presence of benzene and *o-*xylene was related to the toxicity of these compounds. Benzene was the more toxic and less preferred substrate, and *Rhodococcus* cells grew weaker in its presence compared to *o-*xylene. The percentage of strains able to utilize benzene varied from 8% to 35% and was highest among strains of *R. ruber* (Figure 1). Between 2 and 11 times more strains were able to grow in the presence of *o-*xylene. Only *R. opacus* did not use *o-*xylene as a growth substrate, as no *R. opacus* strains grew in the presence of this arene (Figure 1 and Appendix A).

The analysis of the abilities of all tested strains to utilize monoaromatic hydrocarbons confirmed that *o-*xylene was a more preferable substrate for *Rhodococcus* than benzene. The 66 *Rhodococcus* strains from the 117 total strains analyzed grew in the presence of *o-*xylene, which corresponded to 56% *o-*xylene-degrading strains. Only 25 strains from a total of 102 grew in the presence of benzene, which corresponded to 25% benzene-degrading strains (Figure 2a). It was difficult to reveal the substrate preferences of the less represented *Rhodococcus* species (*R. aetherivorans*, *R. cerastii*, *R. corynebacterioides*, *R. globerulus*, *R. jostii*, *R. pyridinivorans*, *R. rhodochrous*, *R. wratislaviensis*, and *R. yannanensis*). However, the strains of these species were able to grow in the presence of benzene and/or *o-*xylene, also preferring *o-*xylene (Appendix A).

The *Rhodococcus* strains were further tested for their ability to use toluene (a benzene homolog with one methyl group) as the sole carbon and energy source. This compound was four times less toxic than benzene and had the intermediate position between benzene and *o-*xylene in terms of its preference as a growth substrate. A total of 39% of strains grew in the presence of toluene (Figure 2a). Among the most represented *Rhodococcus* species, *R. ruber* better utilized toluene: 69% of the tested strains of *R. ruber* (Figure 2c) and only 17% of *R. erythropolis / R. qingshengii* strains (Figure 2b) grew in the presence of this compound.

Furthermore, it was found that *Rhodococcus* bacteria rarely utilized only toluene without the ability to oxidize other monoaromatic hydrocarbons (Figure 2). Only three *R. ruber* strains, IEGM 93, IEGM 232, and IEGM 234, oxidized toluene and did not oxidize benzene and *o-*xylene. Other *Rhodococcus* spp. grew in the presence of toluene only if they were able to grow in the presence of benzene and/or *o-*xylene (Appendix A). The most promising *Rhodococcus* strains, namely *R. aetherivorans* IEGM 1250, *R. erythropolis* IEGM 1232, *R. fascians* IEGM 1233, *R. ruber* IEGM 1121, and *R. ruber* IEGM 1156, were selected, which grew well in the presence of the three tested monoaromatic hydrocarbons (Appendix A).

### 2.3. Degradation of PAHs by Rhodococcus

The 67 *Rhodococcus* strains were screened for their abilities to oxidize naphthalene, the simplest polyaromatic hydrocarbon. It was shown that 41 strains (61%) grew in the presence of naphthalene. Comparing the two most represented species (*R. erythropolis / R. qingshengii* and *R. ruber*), the percentage of naphthalene-degrading strains was similar, constituting 53 and 63%, respectively. All tested *R. opacus* (4) and *R. rhodochrous* (5) strains were able to utilize naphthalene (Figure 3).

The 20 strains growing in the presence of naphthalene were tested for their abilities to use other PAHs. As seen from Figure 4, naphthalene-degrading strains utilized heavier PAHs with between three and five condensed aromatic rings—phenanthrene, anthracene (PAHs with three rings), benzo[a]anthracene (a PAH with four rings), and benzo[a]pyrene (a heavy PAH with five rings). However, the relationship between the intensity of growth and *Rhodococcus* species or hydrocarbon specificity was not statistically significant.

### 2.4. Degradation of Substituted Aromatic Hydrocarbons and Aromatic Heterocycles by Rhodococcus

Several *Rhodococcus* strains were tested for their abilities to use phenol and pyridine. As shown in Table 2, these substrates were poorly oxidized by *Rhodococcus*. Only 16% of the selected strains grew in the presence of phenol, and phenol-oxidizing strains were equally represented among *Rhodococcus* species. However, the percentage of phenol-degrading *R. ruber* strains was higher than that of phenol-degrading *R. erythropolis/R. qingshengii* strains, constituting 19 and 13%, respectively (Table 2). Considering N-heterocyclic pyridine, only two strains out of nine (22%), *R. rhodochrous* IEGM 757 and *R. ruber* IEGM 231, were able to metabolize this compound (Table 2, Appendix A).

*R. ruber* IEGM 231 was tested for aromatic compounds with a nitrogen atom. Methyl- and hydroxy-substituted derivatives of pyridine, a two-ring N-heterocycle quinoline, and the simplest aromatic amine aniline were used as growth substrates. It was shown that *R. ruber* IEGM 231 grew in the presence of all pyridine derivatives and aniline. However, it was not able to use quinoline as the sole carbon and energy source (Figure 5).

Additionally, the ability of *R. ruber* IEGM 231 to use aromatic acids and their derivatives (salicylic acid, phthalic acid isomers, and coumarin as the lactone of *o*-hydroxycinnamic acid) as growth substrates was estimated. It was shown that *R. ruber* IEGM 231 grew in the presence of coumarin but was not able to use salicylic and phthalic acids (Figure 5).

### 2.5. Biodegradation of Toxic Aromatic Compounds by Rhodococcus in Model Soil

The bioremediation potential of *Rhodococcus* bacteria toward aromatic compounds was estimated in experiments with model soil contaminated with PAHs at a high (1 g/kg) concentration. Introducing *Rhodococcus* resulted in a 43% PAH biodegradation within 213 days, which was three times more efficient than in contaminated soil without *Rhodococcus* (Figure 6). Enhanced biodegradation of PAHs was accompanied by a 2.5–10.0-fold increase in the number of hydrocarbon-oxidizing microorganisms. The highest number of hydrocarbon degraders, 2 × 10^8^ colony-forming units (CFU)/g soil, was determined between 35 and 56 days. A significant increase in the bacterial number was also detected in the control soil; however, this value reached no more than 1 × 10^8^ CFU/g. The number of hydrocarbon-oxidizing microorganisms started to decrease after the 56th and 70th days in bioaugmented and control soil. However, it remained at a high ((1.0 ± 0.3) × 10^8^ CFU/g) level in soil with introduced *Rhodococcus* bacteria until the end of the experiment and reduced to almost the initial level ((3.0 ± 0.3) × 10^7^ CFU/g) in the control soil after the 186th day (Figure 6).

In contrast to the dynamics of bacterial cell numbers, the PAH concentration dynamics followed a linear trend, thus demonstrating stable hydrocarbon activity in bioaugmented soil for all 213 days. In the control soil, a slow linear decline in PAH concentration was also revealed, but it did not change significantly after the 70th day (Figure 6). The removal of PAHs both in the soil with introduced *Rhodococcus* cells and control was related to a decrease in the concentrations of individual, mainly light PAHs—naphthalene and acenaphthene. These hydrocarbons were evaporated and easily biodegraded because of their relatively high water solubility compared to other PAHs [34,39].

### 2.6. Analysis of Rhodococcus Genes Involved in the Biodegradation of Aromatic Compounds

Various oxidoreductases and hydrolases were revealed in sequenced and annotated genomes of studied *Rhodococcus* strains. The total numbers of oxidizing enzymes, such as dioxygenases, monooxygenases, multicopper polyphenol oxidases, dehydrogenases, and hydrolases, varied in the ranges of 23–69, 45–97, 1–4, 256–731, and 96–229, respectively. The highest numbers of putative aromatic-degrading enzymes were detected in the genome of *R. opacus* IEGM 249 (Table 3 and Appendix A).

Dioxygenases are key enzymes of aromatic compound metabolism. They incorporate two oxygen atoms into the central metabolites (catechol, protocatechuate, gentisate, homogentisate, etc.), thus resulting in ring cleavage [18]. In the genomes of the studied *Rhodococcus* strains, dioxygenases for ortho (intradiol) and meta (extradiol) ring cleavage were found. They included catechol 1,2- and protocatechuate 3,4-dioxygenases for intradiol oxidation, and catechol 2,3-, gentisate 1,2-, homogentisate 1,2-, and other extradiol dioxygenases for extradiol oxidation. The preferred pathway for catechol cleavage was, apparently, ortho-oxidation as catechol 1,2-dioxygenases were revealed in all studied genomes. Catechol 2,3-dioxygenases were found only in *R. opacus* IEGM 249, *R. pyridinivorans* IEGM 1137, and *R. ruber* IEGM 231 genomes (Table 3 and Appendix A). Other enzymes that could lead to the multistep formation of key metabolites from complex aromatic compounds with several rings and various functional groups [18] included phenol monooxygenases, 4-hydroxybenzoate 3-, 3-hydroxybenzoate 6-, and salicylate 1-hydroxylases, and vanillate 3-O demethylases (Table 3).

Among specific genes coded for the enzymes oxidizing benzene and its homologs, one gene annotated as a benzene 1,2-dioxygenase was found in the genome of *R. ruber* IEGM 231. A benzoyl-CoA oxygenase/reductase gene providing dearomatization [18] was revealed in genomes of *R. opacus* IEGM 249, *R. rhodochrous* IEGM 107, and IEGM 1360. Another enzyme for dearomatization was phenylacetyl-CoA oxygenase/reductase found in sequenced *R. erythropolis/R. qingshengii* strains. Phenol monooxygenases [3] found in *R. ruber* IEGM 231 and phenol hydrolases found in *R. opacus* IEGM 249 and *R. pyridinivorans* IEGM 1137 could participate in the degradation of phenol. Among enzymes probably participating in the biodegradation of aromatic compounds with nitrogen atoms (aniline, pyridine, and its derivatives), genes coded for anthranilate 1,2- and 3-hydroxy anthranilate 3,4-dioxygenases, maleamate isomerases, amidases including formamidases, maleamate amidohydrolases, aldehyde dehydrogenases, and succinate-semialdehyde dehydrogenases [9,18,20,23] but not aniline, 2-aminophenol 1,6-, indole 3-acetate dioxygenases, 6-hydroxynicotinate 3-monooxygenases and 2-aminobenzoyl-CoA monooxygenases/reductases [18,40] were revealed (Table 3 and Appendix A). This spectrum of putative enzymes suggests the biodegradation of pyridine via hydrogenation followed by cleavage of the C2-C3 double bond [9]. Among the genes of coumarin biodegradation, those coded for fumarylacetoacetate and hydroxymuconic semialdehyde hydrolases, flavin-binding hydroxylases, and extradiol dioxygenases but not 3-(3-hydroxy-phenyl)propionate hydroxylases [25] were detected in *Rhodococcus* genomes. No gene evidently responsible for the biodegradation of phthalates, such as phthalate, terephthalate and isophthalate dioxygenases, 3,4-dihydroxy-3,4-dihydrophthalate and 4,5-dihydro-4,5-dihydroxyphthalate dehydrogenases, and 4,5-dihydroxyphthalate and 3,4-dihydroxyphthalate decarboxylases [41], were revealed in the sequenced genomes. Only the *R. opacus* strain IEGM 249 had one gene of phthalate 3,4-dioxygenase but no genes coded for other enzymes of phthalate biodegradation (Table 3 and Appendix A).

## 3. Discussion

Based on the metabolic abilities of 133 *Rhodococcus* strains from the Regional Specialized Collection of Alkanotrophic Microorganisms, common regularities of aromatic compound biodegradation by *Rhodococcus* bacteria were revealed and discussed in terms of the toxicity and putative functional genes. According to the preference for *Rhodococcus* cells, the studied aromatic substrates can be arranged (from more to less preferrable) as PAHs (61% of strains used naphthalene, and all tested naphthalene-degrading strains grew in the presence of PAHs with between three and five aromatic rings) > *o-*xylene (56% of strains used) > toluene (39% of strains used) > benzene (25% of strains used) > pyridine (22% of strains used) > phenol (16% of strains used). This preference row reflected the toxicity of aromatic hydrocarbons (Table 1), which increased from PAHs to benzene. Interestingly, all PAHs used, regardless of the number of aromatic rings, molecular weight, and even water solubility [33], were equally preferred for *Rhodococcus* cells (Figure 4). A similar effect was found earlier for *Rhodococcus*, and it depended on the strain specificities and cell adhesive activities toward PAH crystals [34]. However, the polar-substituted derivatives of benzene, pyridine, and phenol did not fully correspond to the revealed dependence. These less-toxic compounds were degraded more poorly compared to highly toxic benzene and toluene, which could be explained by the specific enzymatic mechanisms of their degradation.

It was expected that this study would reveal a certain species specificity of the metabolic profiles (spectra of oxidized aromatic compounds) of the studied *Rhodococcus* strains. However, no strict correlation between species and the ability of *Rhodococcus* strains to metabolize specific aromatic substrates was detected. Strains able to grow in the presence of benzene, toluene, *o-*xylene, naphthalene, pyridine, or phenol and not using these compounds were found among different *Rhodococcus* species. Similarly, strains of *R. erythropolis/R. qingshegii*, *R. opacus*, *R. rhodochrous*, and *R. ruber* species grew in the presence of PAHs (Figure 4). Nevertheless, some species specificities were revealed: (i) *R. opacus* did not use *o-*xylene, and (ii) *R. ruber* somewhat better-metabolized aromatic compounds, especially in comparison with the most represented *R. erythropolis/R. qingshegii*. The latter was confirmed by the highest percentage of *R. ruber* strains growing in the presence of benzene and the 1.5 and 4.0 higher percentage of strains oxidizing phenol and toluene in comparison with *R. erythropolis/R. qingshengii*. Moreover, the ability of *R. ruber* to oxidize pyridine (pyridine-degrading strains were not found among *R. erythropolis/R. qingshegii* strains) and the ability of the reference strain *R. ruber* IEGM 231 to oxidize most of the tested aromatic pollutants indicate a high biodegradation potential of this *Rhodococcus* species. The metabolic potential of *R. rhodochrous* was close to that of *R. ruber* as all tested *R. rhodochrous* strains grew in the presence of PAHs and phenol, and *R. rhodochrous* IEGM 757 was able to use pyridine. Possible explanations for the inability of *R. opacus* to use *o-*xylene, a relatively less toxic compound, are the absence of specific oxidizing enzymes, the low affinity of benzene/toluene-degrading enzymes to *o-*xylene, or the lack of specific transporting systems. In fact, *R. opacus* poorly metabolized monoaromatic hydrocarbons. Even considering benzene and toluene, only three strains of this species grew in the presence of these compounds (Appendix A). Genes annotated for benzene/toluene monooxygenases or dioxygenases were not found in the genome of *R. opacus* IEGM 249 (Appendix A), which did not utilize monoaromatic hydrocarbons (Appendix A). It should be noted, however, that benzene/toluene mono- and dioxygenases were not detected in the most studied *Rhodococcus* genomes (Appendix A). Nevertheless, *R. opacus* was recognized as a promising degrader of toxic organic pollutants as the highest numbers of oxidoreductases were revealed in the genome of *R. opacus* IEGM 249 (Table 3).

No dependence was found between the abilities of *Rhodococcus* strains to oxidize aromatic compounds, the spectra of oxidized aromatic substances, and the source of strain isolation. The following strains were selected as the most promising degraders: *R. aetherivorans* IEGM 1250 and *R. erythropolis* IEGM 1232 (used all three monoaromatic hydrocarbons—benzene, toluene, and *o*-xylene); *R. ruber* IEGM 1121 and IEGM 1156 (used benzene, toluene, *o-*xylene, PAHs, and weakly grew in the presence of phenol); *R. erythropolis* IEGM 190, IEGM 201, and IEGM 265 (used phenol); *R. rhodochrous* IEGM 1161 (used phenol and also naphthalene); *R. erythropolis* IEGM 788 and *R. rhodochrous* IEGM 1162 (used xylene, naphthalene, and phenol); *R. ruber* IEGM 1122 (used benzene, toluene, *o-*xylene, naphthalene, and phenol); *R. rhodochrous* IEGM 757 (used pyridine); *R. ruber* IEGM 231 (a well-characterized strain with high metabolic potential, able to use benzene, toluene, PAHs, pyridine and its derivatives, aniline, and coumarin). The *R. fascians* strain IEGM 1233 degraded all the studied monoaromatic hydrocarbons; however, as it represented a phytopathogenic species [42], we did not consider it to be an appropriate biotechnology agent.

It was hypothesized that microorganisms from anthropogenically disturbed and contaminated ecosystems had developed diverse metabolic capacities and multiple resistance to stress factors [27,28]. Among the selected 14 most active biodegraders, 7 strains were isolated from pristine environments, and the other 7 were isolated from disturbed sites, either polluted or located near pollution sources (Appendix A). In fact, the impact of the isolation source on the biodegradation abilities of *Rhodococcus* was difficult to identify. For example, when strains were isolated within the city, analysis suggested some level of chemical contamination, even if no pollutants were defined and the sites were assumed to be pristine.

Special attention in this study was paid to the analysis of putative biodegradation genes. For this, 10 sequenced genomes of *Rhodococcus* strains from the IEGM Collection were annotated and searched for relevant functional genes. The obtained results proposed enzymatic mechanisms of aromatic compound biodegradation and predicted metabolic activities of particular *Rhodococcus* species. Catechol dioxygenases, key enzymes of arene biodegradation pathways, were found in all of the studied genomes of IEGM strains. This confirmed the metabolic ability of *Rhodococcus* to degrade molecules with aromatic structures as well as the preferred pathway for further catechol degradation through ortho-cleavage (Table 3). Considering the genes of the initial steps of aromatic biodegradation, they could be assumed as follows. One benzene 1,2-dioxygenase and three phenol monooxygenases were revealed in the *R. ruber* IEGM 231 genome, and phenol hydrolases (one gene in each strain) were revealed in the genomes of *R. opacus* IEGM 249 and *R. pyridinivorans* IEGM 1137 (Table 3). These enzymes can provide primary oxidation of BTEX substrates (benzene, toluene, and *o-*xylene) and phenol, respectively. The highest number of specific enzymes in *Rhodococcus* genomes was revealed for the degradation of pyridine, its derivatives, and coumarin. The ability of *R. ruber* IEGM 231 to grow in the presence of 2- and 4-hydroxypyridines verified that rhodococci metabolized pyridine completely as these compounds were described as frequent end products in pyridine biodegradation [24]. In four *Rhodococcus* genomes, a gene coded for salicylate 1-hydroxylase, presumably responsible for salicylic acid assimilation, was revealed. This gene was not found in the *R. ruber* IEGM 231 genome (Table 3), which is a possible reason for the inability of this strain to metabolize salicylate (Figure 5). Interestingly, salicylate 5-hydroxylases were not found in the sequenced genomes (Appendix A). Salicylic acid, along with phthalic acids, is known as an intermediate in PAH biodegradation [43,44]. The revealed genes indicated that the utilization of PAHs by *Rhodococcus* cells is performed, apparently, via catechol, but not the gentisate pathway [18].

It was surprising that no genes recognized as naphthalene or other PAH dioxygenases were detected in *Rhodococcus* genomes. Naphthalene dioxygenases were previously described for *Rhodococcus* spp. in relation to their structural, functional, and thermostability features [43,45,46,47]. The explanation could be that dioxygenases belong to a diverse class of Rieske non-heme iron oxygenases with a broad substrate specificity. They include, among others, phthalate, biphenyl, benzoate, toluene dioxygenases, and some monooxygenases [45,48]. Genes encoding dioxygenases can be annotated as extradiol dioxygenases, unspecific dioxygenases, or unspecific oxidases/oxidoreductases. The low (29–33%) similarity of naphthalene dioxygenase and, in contrast, high (77–81%) similarity of nitrobenzene dioxygenase subunits NarAa and NarAb from *Rhodococcus* sp. strain NCIMB12038 with the corresponding subunits from *Pseudomonas* [45] demonstrated a high diversity of naphthalene enzymes and possible misleading annotations. More detailed analysis of dioxygenase genes using multiple alignments and experimental confirmation of gene functions are required. Moreover, naphthalene-degrading operons can be located on plasmids, the absence of which may explain the inability of some *Rhodococcus* strains to grow in the presence of this compound.

The variety of oxidizing enzymes with a broad substrate specificity found in *Rhodococcus* genomes determines the versatile and widely varying catabolic abilities of the studied strains. Hydrolases seemed to be no less important enzymes for the biodegradation of aromatic compounds by *Rhodococcus* bacteria than dioxygenases. Numerous genes coded for hydrolases (from 96 up to 229) were found in all studied *Rhodococcus* genomes (Table 3). They can participate in the degradation of PAH molecules [30]. Among hydrolases, formamidases and other amidases were detected. These enzymes, along with maleamate amidohydrolase found in the genome of *R. ruber* IEGM 231 (Appendix A), can participate in the biodegradation of pyridine and its derivatives [9,18]. In addition to dioxygenases and hydrolases, putative aromatic biodegradation enzymes also included monooxygenases, especially cytochrome P450 oxidases, and multicopper polyphenol oxidases, or laccases. The latter can cause the oxidation of PAHs, phenolics, and other toxic pollutants [49,50], thereby allowing the growth of *Rhodococcus* in the presence of polyaromatics and phenol, even if specific enzymes are not annotated. Multicopper polyphenol oxidases were found in all the studied genomes of *Rhodococcus* spp., mainly in one copy, while four genes were detected in *R. ruber* IEGM 231 (Table 3).

Some contradictory data were obtained for the biodegradation of phthalic acids aniline and quinoline. The reference strain *R. ruber* IEGM 231 did not grow in the presence of phthalic acid isomers, which was in agreement with the genome annotation: no genes coded for phthalic acid biodegradation enzymes were found in the genome of this strain or in the genomes of other *Rhodococcus* strains. Only one phthalate 3,4-dioxygenase gene was recognized in the genome of *R. opacus* IEGM 249 (Appendix A). However, it is known that *Rhodococcus* bacteria can degrade phthalates and their esters, and phthalate 3,4-dioxygenases coded by the *pht* genes participate in these metabolic processes [31,32,41]. Phthalate operons are located on plasmids [32,41], and the lack of plasmids can explain the inability of the IEGM 231 strain to metabolize phthalic acids. Another substrate, which was not used by *R. ruber* IEGM 231, was quinoline. In the genome of this strain, as well as in other *Rhodococcus* genomes, genes coding for nitrite and nitrate reductases were revealed (Appendix A). These enzymes could perform the degradation of quinoline via the 8-hydroxycoumarin pathway [20,23]. The inability of IEGM 231 to use quinoline can be due to the lack of other key enzymes or transporting proteins, but apparently not to quinoline toxicity, which was lower than benzene and toluene were. In contrast, *R. ruber* IEGM 231 grew in the presence of aniline, but no specific genes coded for the enzymes of the first steps of aniline biodegradation, such as aniline, 2-aminophenol 1,6-, indole 3-acetate dioxygenases, 6-hydroxynicotinate 3-monooxygenases, and 2-aminobenzoyl-CoA monooxygenases/reductases, were found in the genome of this strain. Enzymes with a broad substrate specificity (dioxygenases, for example) or enzymes of putative pyridine oxidation pathways (anthranilate 1,2-dioxygenase, amidases, or maleamate amidohydrolase) can participate in the degradation of aniline by *R. ruber* IEGM 231.

The metabolic properties of *Rhodococcus* strains from the IEGM Collection toward aromatic compounds were tested against heavy (1 g/kg) PAH contamination in a model soil. The obtained results showed the suitability of introducing *Rhodococcus* to remove aromatic hydrocarbons from soil. The percentage of removed hydrocarbons was three times higher in soil with introduced *Rhodococcus* bacteria compared to control soil and constituted 43% PAH removal over 213 days (Figure 6). To improve biodegradation, *Rhodococcus* cells were immobilized onto a sawdust-based carrier [51]. Immobilized cells maintained high PAH-oxidizing activity throughout the experiment. This was seen from the constant linear decline in PAH concentration, even after the total number of hydrocarbon-oxidizing microorganisms was decreased. The growth of PAH degraders within the first 35 days was seemingly related to easily biodegradable aromatic components, such as light PAHs or PAHs not sorbed on solid particles (Figure 6). After the depletion of these components, the abundance of hydrocarbon-oxidizing bacteria decreased, and the process of PAH biodegradation stopped in the control but continued in soil with introduced *Rhodococcus* cells. To further demonstrate the metabolic capabilities of *Rhodococcus* bacteria toward aromatic substances, their high oxidizing activities toward complex aromatic compounds, such as (RS)-2-(4-(2-methylpropyl)phenyl)propanoic and [2-(2,6-dichloroanilino)phenyl]acetic acids, should be mentioned (Table 4). These complex aromatic acids were acting substances in non-steroidal anti-inflammatory drugs, and *Rhodococcus* strains from the IEGM Collection degraded 100% of these compounds at high concentrations within 6 days (Table 4).

## 4. Materials and Methods

### 4.1. Bacterial Strains and Culture Conditions

The 133 pure identified cultures of *Rhodococcus* spp. from the Regional Specialized Collection of Alkanotrophic Microorganisms of the Institute of Ecology and Genetics of Microorganisms, Perm, Russia (IEGM; www.iegmcol.ru; WFCC 285; UNU/CKP 73559/480868) belonging to *R. aetherivorans* (1 strain), *R. erythropolis* (41 strains), *R. qingshengii* (7 strains), *R. cerastii* (1 strain), *R. corynebacterioides* (2 strains), *R. fascians* (12 strains), *R. globerulus* (2 strains), *R. jostii* (3 strains), *R. opacus* (17 strains), *R. pyridinivorans* (2 strains), *R. rhodochrous* (11 strains), *R. ruber* (32 strains), *R. wratislaviensis* (1 strain), and *R. yunnanensis* (1 strain) were used in this study. Strains were isolated from various pristine and contaminated sources (Appendix A). Currently, *R. qingshengii* is a synonym of *R. erythropolis* (https://lpsn.dsmz.de/species/rhodococcus-qingshengii, accessed on 2 March 2023) [54]; therefore, the results obtained for strains of these two species are presented in this work as those obtained for the *R. erythropolis/R. qingshengii* group.

*Rhodococcus* cells were recovered from lyophilized cultures and then grown in Erlenmeyer flasks containing 100 mL of Luria–Bertani broth (LB) (Sigma-Aldrich, Burlington, VT, USA) on an orbital shaker (160 rpm) at 28 °C for 28–30 h. Cells were washed twice and resuspended in 0.5% NaCl to OD_600 nm_ = 1.0 (1 × 10^8^ CFU/mL). The use of standardly prepared cell suspensions provided the same initial conditions in all experiments.

### 4.2. Tested Aromatic Compounds

The metabolic abilities of *Rhodococcus* strains were estimated toward 23 aromatic compounds, including 3 monocyclic aromatic hydrocarbons, 5 PAHs, 1 hydroxy substituted benzene, 1 aromatic amine, 8 N-heterocyclic aromatic compounds, 4 aromatic acids, and 1 aromatic lactone. These compounds are listed in Table 5. All substances were dissolved in water or in polar solvents, such as dimethylsulfoxide (DMSO), acetone, or 70% ethanol (Table 5) at a concentration of 600 mM. Sometimes, the solutions were heated in a water bath at 70 °C for better dissolution (Table 5). The aromatic compounds and organic solvents had ≥97% purity and were purchased from Sigma-Aldrich. Pre-sterilized glass- and plasticware were used for the preparation of aromatic compound solutions, and water solutions of pyridine and its derivatives were sterilized by filtering through nitrocellulose filters with 0.22 μm pores (Merck Millipore, Burlington, VT, USA).

### 4.3. Toxicity Tests

Minimal inhibitory concentrations (MICs) of the used aromatic compounds were determined for the reference strain *R. ruber* IEGM 231. In polystyrene 96-well microplates (Medpolymer, St Petersburg, Russia), 150 μL of sterile LB was mixed with 30 μL working solutions of aromatic compounds to obtain a starting concentration of 100 mM. Then, a series of two-fold dilutions in the range of concentrations between 0.05 and 100.00 mM were prepared. Microplates were stored at room temperature for 24 h to allow the evaporation of solvents. Measures of 10 μL of a standard *R. ruber* IEGM 231 cell suspension were added to microplates to obtain a final concentration of 1 × 10^7^ CFU/mL. Inoculated microplates were incubated in a Titramax 1000 incubator (Heidolph Instruments, Schwabach, Germany) at 600 min^−1^ and 28 °C for 72 h. The viability of cells was estimated using staining with iodonitrotetrazolium violet (INT) purchased from Sigma-Aldrich. For this, measures of 45 μL of 0.2% (*w*/*w*) INT solution in water were added to microplates. The appearance of a red-violet color after 2 h of staining was evidence of the presence of viable respirating cells [55]. LB with *R. ruber* IEGM 231 cells without aromatic substrates was the biotic control. LB with cells and solvents without dissolved aromatic compounds were used to estimate the influence of solvents on the growth of *Rhodococcus*. LBs with aromatic compounds without *R. ruber* IEGM 231 cells were used as abiotic controls.

### 4.4. Growth Experiments

To estimate the abilities of *Rhodococcus* strains to use benzene, toluene, *o-*xylene, naphthalene, pyridine, and phenol as growth substrates, bacterial cells were grown on mineral agar K containing (g/L): KH_2_PO_4_—1.0, K_2_HPO_4_—1.0, NaCl—1.0, KNO_3_—1.0, MgSO_4_—0.2, FeCl_3_—0.02, CaCl_2_—0.02, and agar—15.0 (http://www.iegmcol.ru/medium/med08.htmL, accessed on 2 March 2023). The 200 μL diluted *Rhodococcus* cell suspensions with a concentration of 1·10^7^ CFU/mL were used for the inoculation of Petri dishes with agar K. Suspensions were evenly distributed on the agar surface with a spatula to obtain a bacterial lawn when growing. Liquid substrates (monoaromatic hydrocarbons, pyridine, and phenol) at a volume of 200 μL were added to the slot with a diameter of 1 cm made in the center of the Petri dishes with agar K. A measure of 200 mg of naphthalene was placed under each lid of the inoculated Petri dishes, and the dishes were inverted. All inoculated Petri dishes were incubated at 28 °C for 5 days.

To estimate the abilities of *Rhodococcus* strains to use anthracene, phenanthrene, benzo[a]anthracene, benzo[a]pyrene, salicylic acid, phthalic acid isomers, 2-,3-,4-picolines, 2,6-lutidine, 2-,4-hydroxypyridines, quinoline, and coumarin, bacterial cells were inoculated into a mineral medium “*Rhodococcus-*surfactant” (RS) in 96-well polystyrene microplates. Naphthalene was also used in these experiments as a reference PAH. Solutions of tested compounds prepared as described in Section 4.2 were added to 100 μL of the RS medium to produce final concentrations two times lower than those of MIC. The RS medium contained (g/L): K_2_HPO_4_—2.0, KH_2_PO_4_—2.0, KNO_3_—1.0, (NH_4_)_2_SO_4_—2.0, NaCl—1.0, MgSO_4_—0.2, CaCl_2_—0.02, and a trace element solution—1 mL/L (www.iegm.ru/iegmcol/medium/med11.html, accessed on 2 March 2023). Then, 10 μL measures of a standard *Rhodococcus* cell suspension were added, and microplates were incubated in the Titramax 1000 incubator at 300 min^−1^ at 28 °C for 48 h.

The used salts and D-glucose had ≥97% purity and were purchased from Sigma-Aldrich, and the agar was ultrapure BD Difco^TM^, purchased from Thermo Fisher Scientific (Waltham, MA, USA). Uninoculated mineral medium supplemented with the studied aromatic compounds was used as an abiotic control; LB agar or the medium RS with 25 mM D-glucose inoculated with *Rhodococcus* bacteria was the biotic control; mineral medium without aromatic substrates inoculated with *Rhodococcus* bacteria was the control for oligotrophic growth. The usage of ultrapure Difco agar guaranteed that no residual growth was detected as a result of the consumption of trace concentrations of organic molecules by *Rhodococcus* [26,27,35]. Only the results of growth experiments with the confirmed growth of *Rhodococcus* cells in the biotic control and the absence of oligotrophic growth are shown and discussed in this work.

### 4.5. Model Soil Experiments

The model soil consisted of 50% sand, 30% clay, and 20% clean garden soil. All components of the soil mixture were first dried at 80 °C and then screened using a 1 mm mesh. The soil was inoculated with acetone-dissolved PAHs. Individual PAHs were present in the following concentrations (g/kg dry soil): naphthalene—0.2, acenaphthene (>97% purity, Sigma-Aldrich)—0.2, anthracene—0.2, phenanthrene—0.2, benzo[a]anthracene—0.1, and benzo[a]pyrene—0.1. Biodegradation of the PAH mixture was performed in flat trays containing 600 g of soil. The *R. erythropolis* IEGM 708 and *R. ruber* IEGM 327 strains [56] were used in this experiment in equal amounts. To stabilize the metabolic activity of *Rhodococcus* cells for an extended period, they were immobilized onto pine sawdust hydrophobized using 5% *Rhodococcus* surfactant, as described previously [51]. The catalytic activity of immobilized cells was measured as 98 ± 12 mg of degraded naphthalene/(L·h). The obtained biocatalyst was introduced into the contaminated soil at a 1:6 ratio (*w*/*w*); the amount of the inoculum was 4 × 10^8^ cells/g of soil. The soil was mixed and moistened regularly to maintain 20% humidity. The experiment was performed at room temperature for 213 days. Contaminated soil without *Rhodococcus* cells introduced was used as a control.

Residual hydrocarbons were determined by gas chromatography with mass spectrometry after extraction with chloroform (GC-MS grade). An Agilent 6890 N chromatograph equipped with an Agilent MSD 5973 N quadrupole detector (Agilent Technologies, Santa-Clara, CA, USA) was used. A volume of 1 μL of each extract was introduced into an injection port held at 250 °C. The initial oven temperature was 40 °C for 5 min followed by a heating rate of 12 °C/min up to 300 °C, and held at this temperature for 10 min. Separation was achieved using a 30 m HP-5MS column with an internal diameter of 0.25 mm and film thickness of 0.25 M (Agilent Technologies, Santa-Clara, CA, USA) maintained at a constant flow of 1 mL/min of helium. The number of hydrocarbon-oxidizing microorganisms was counted as the number of colony-forming units grown on agar K with naphthalene.

### 4.6. Bioinformatical and Statistical Analysis

Draft genome sequences of *R. erythropolis* IEGM 267 (DDBJ/ENA/GenBank accession no. MRBQ01000001-MRBQ01000231), IEGM 746 (JAJNDF010000001-JAJNDF010000078), IEGM 1189 (JAPWIG010000001-JAPWIG010000056), IEGM 1359 (JAJNCL010000001-JAJNCL010000048), *R. opacus* IEGM 249 (JAPWIS010000001-JAPWIS010000138), *R. pyridinivorans* IEGM 1137 (JAPWII010000001-JAPWII010000087), *R. rhodochrous* IEGM 107 (JAJNCP010000001-JAJNCP010000118), IEGM 757 (JAJNCO010000001-JAJNCO010000163), IEGM 1360 (JAJNCN010000001-JAJNCN010000105), and *R. ruber* IEGM 231 (CCSD01000001-CCSD01000115) were used to search for functional genes coded for enzymes participating in biodegradation of aromatic compounds. The annotation of coding sequences in the genomes was performed using the RAST 2.0 annotation scheme RASTtk [57].

All experiments were performed in 3–8 replicates. Statistical analysis including the determination of the data type distribution and calculation of means ± standard deviations was performed using Statistica (data analysis software system) version 13, TIBCO Software Inc. (2018). Differences were considered statistically significant at *p* < 0.05.

## 5. Conclusions

The results presented in this work were obtained in the course of long-term and comprehensive studies of the abilities of *Rhodococcus* bacteria to metabolize aromatic compounds. They are part of a larger study of the biological and functional properties of this group of bacteria performed using the bioresources of the Regional Specialized Collection of Alkanotrophic Microorganisms. *Rhodococcus* strains deposited in the collection were isolated from various pristine and anthropogenically disturbed ecosystems, identified and well characterized. Moreover, the collection is constantly updated with freshly isolated cultures. In this work, the metabolic properties of the 133 collection strains belonging to various species of the genus *Rhodococcus* were determined. We assume that the obtained results were representative of the whole genus. No strict correlations were revealed between the abilities of *Rhodococcus* isolates to oxidize aromatic compounds, spectra of oxidized aromatic substances, and species. However, *R. rhodochrous* and *R. ruber* somewhat better oxidized various aromatic compounds, and *R. opacus* poorly metabolized monoaromatic hydrocarbons and did not use *o-*xylene. No dependence was revealed between the metabolic abilities of the studied strains toward aromatic compounds and the source of strain isolation. The toxicity of aromatic substances was not the only factor determining their bio-oxidation. The diversity of genes coded for degrading enzymes with a broad substrate specificity (mono- and dioxygenases, laccases, and hydrolases) was considered to be the basis for the versatile and widely varying catabolic abilities of *Rhodococcus* bacteria. At the same time, the analysis of functional annotations of 10 *Rhodococcus* genomes revealed that there is still no sufficient information on the enzyme mechanisms of aromatic compound degradation by *Rhodococcus*. Individual *Rhodococcus* strains may have no specific genes and be unable to metabolize some aromatics, even if these activities have previously been shown for other members of the genus. In this study, metabolic pathways for mono- and polyaromatic hydrocarbons, phenol, and nitrogen-containing aromatic compounds in *Rhodococcus*, proceeding through the formation of catechol as a key metabolite with its following ortho-cleavage or via the hydrogenation of aromatic rings, were verified. However, further experimental verification of particular gene functions is required.

Information on the IEGM Collection *Rhodococcus* strains can be used on an operational basis to construct biopreparations for emergency spills of ecopollutants, including aromatic pollutants, alone or simultaneously presented in the biotope with other xenobiotics. The developed biopreparations can be used in the bioremediation of soils contaminated with aromatics and requested by companies working in the area of ecobiotechnology. In this work, some promising biodegraders were selected, which metabolized a high diversity of aromatic pollutants and degraded the most recalcitrant substances. In model experiments, the high efficiency of the introduction of immobilized *Rhodococcus* cells for the remediation of contaminated soil was shown, which led to the 43% removal of PAHs for 213 days at an initial PAH concentration of 1 g/kg.

## Figures and Tables

**Figure 1 molecules-28-02393-f001:**
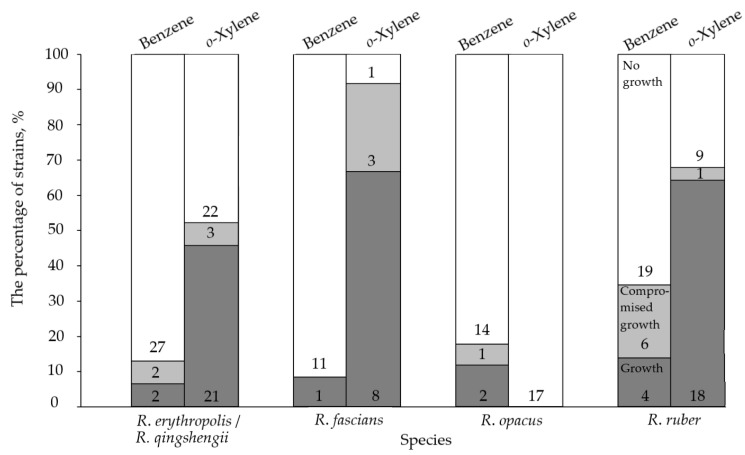
Growth of strains of the most represented *Rhodococcus* species in the presence of benzene and *o-*xylene. The numbers in the columns show the number of strains able to use benzene or *o-*xylene as growth substrates (shaded columns) or unable to grow in their presence (white columns). Compromised growth means weak growth (the growth that was not detected in all replicates).

**Figure 2 molecules-28-02393-f002:**
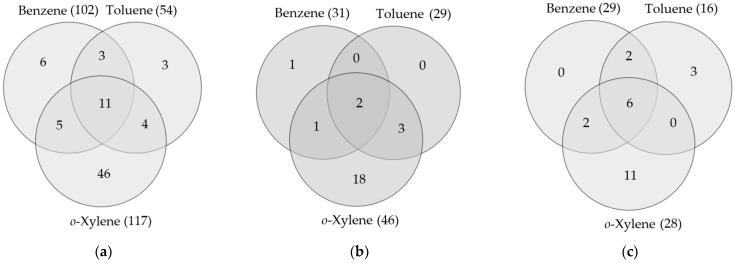
Substrate specificities of *Rhodococcus* strains toward monoaromatic hydrocarbons. Total numbers of strains tested toward specific hydrocarbons are shown in brackets. Numbers of strains growing only in the presence of benzene, toluene, or *o-*xylene are shown in non-overlapping areas of circles. The numbers of strains able to use two or all three hydrocarbons as the sole carbon and energy sources are shown in overlapping areas of circles. (**a**) *Rhodococcus* spp. (**b**) *R. erythropolis/R. gingshengii* strains. (**c**) *R. ruber* strains.

**Figure 3 molecules-28-02393-f003:**
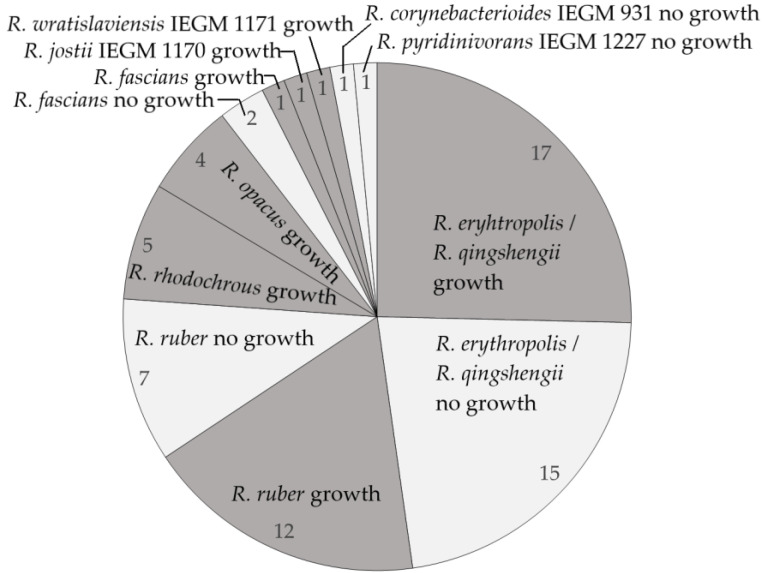
Degradation of naphthalene by *Rhodococcus* spp. Numbers show numbers of strains growing (shaded segments) or not growing (clear segments) in the presence of naphthalene used as the sole carbon and energy source.

**Figure 4 molecules-28-02393-f004:**
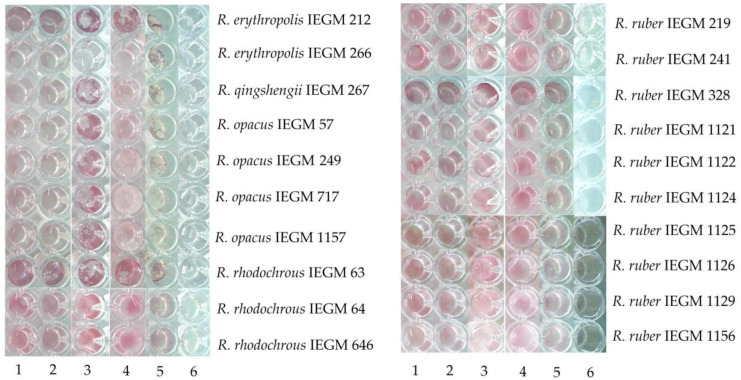
Growth of *Rhodococcus* spp. in the mineral medium RS with 25 mM PAHs: 1—naphthalene, 2—phenanthrene, 3—anthracene, 4—benzo[a]anthracene, 5—benzo[a]pyrene, 6—control for oligotrophic growth (RS without growth substrates). Cells are stained with 0.2% INT. The appearance of red-purple color is evidence of the presence of viable and metabolically active cells.

**Figure 5 molecules-28-02393-f005:**
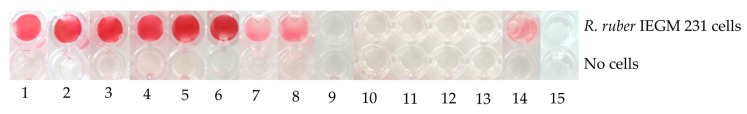
Growth of *R. ruber* IEGM 231 in the presence of aniline, substituted pyridine derivatives, aromatic acids, and coumarin determined with INT staining. Substrates: 1—25.0 mM D-glucose (biotic control); 2—12.5 mM aniline; 3—25.0 mM 2-picoline; 4—25.0 mM 3-picoline; 5—25.0 mM 4-picoline; 6—25.0 mM 2,6-lutidine; 7—25.0 mM 2-hydroxypiridine; 8—25.0 mM 4-hydroxipyridine; 9—3.2 mM quinoline; 10—25.0 mM *o-*phthalic acid; 11—0.1 mM *m-*phthalic acid; 12—0.1 mM *p-*phthalic acid; 13—25.0 mM salicylic acid; 14—25.0 mM coumarin; 15—the mineral medium RS without substrate supplementation (control for oligotrophic growth); no cells—the mineral medium RS with aromatic compounds without *R. ruber* IEGM 231 cells (abiotic control).

**Figure 6 molecules-28-02393-f006:**
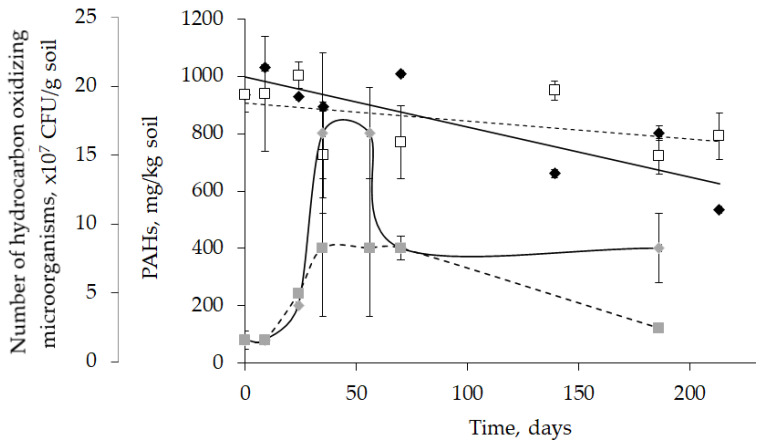
Dynamics of PAH concentration (two upper lines) and number of hydrocarbon-oxidizing microorganisms (two lower graphics) in model soil. Continuous lines correspond to model soil with introduced *Rhodococcus* cells, and dotted lines correspond to control soil without introduced *Rhodococcus* cells. Symbols: black diamonds—concentration of PAHs in soil with *Rhodococcus*; grey diamonds—number of hydrocarbon-oxidizing microorganisms in soil with *Rhodococcus*; white empty squares—concentration of PAHs in control; grey squares—number of hydrocarbon-oxidizing microorganisms in control.

**Table 1 molecules-28-02393-t001:** Minimal inhibitory concentrations of aromatic compounds for *R. ruber* IEGM 231.

Aromatic Compound	MIC, мM	Aromatic Compound	MIC, мM
Benzene	0.2	Phenol	25.0
Toluene	0.8	Aniline	25.0
*o-*Xylene	25.0	Pyridine	50.0
Naphthalene	50.0	2-Picoline	50.0
Anthracene	50.0	3-Picoline	50.0
Phenanthrene	50.0	4-Picoline	50.0
Benzo[a]anthracene	50.0	2,6-Lutidine	50.0
Benzo[a]pyrene	50.0	2-Hydroxipyridine	50.0
*o-*Phthalic acid	50.0	4-Hydroxipyridine	50.0
*m-*Phthalic acid	0.2	Quinoline	6.3
*p-*Phthalic acid	0.2	Coumarin	50.0
Salicylic acid	50.0		

**Table 2 molecules-28-02393-t002:** Growth of selected *Rhodococcus* strains in the presence of phenol and pyridine.

Species	Number of Strains		
	Total	Growing	Not Growing
Phenol			
*R. erythropolis/R. qingshengii*	30	4	26
*R. corynebacterioides*	1 ^1^	0	1
*R. fascians*	4	0	4
*R. jostii*	1 ^1^	0	1
*R. opacus*	1 ^1^	0	1
*R. pyridinivorans*	1 ^1^	0	1
*R. rhodochrous*	2	2	0
*R. ruber*	16	3	13
*R. wratislaviensis*	1	0	1
Total	57	9	1
Pyridine			
*R. aetherivorans*	1	0	1
*R. erythropolis/R. qingshengii*	2	0	2
*R. jostii*	1 ^2^	0	1
*R. pyridinivorans*	1 ^2^	0	1
*R. rhodochrous*	3	1	2
*R. ruber*	2	1	1
Total	9	2	7

^1^*R. corynebacterioides* IEGM 931, *R. jostii* IEGM 1170, *R. opacus* IEGM 1157, and *R. pyridinivorans* IEGM 1227 were tested. ^2^
*R. jostii* IEGM 1193 and *R. pyridinivorans* IEGM 1137 were tested.

**Table 3 molecules-28-02393-t003:** Numbers of genes coded for putative enzymes of aromatic compound biodegradation by *Rhodococcus* spp.

Strains	*R. qingshengii* IEGM 267	*R. qingshengii* IEGM 746	*R. erythropolis* IEGM 1189	*R. qingshengii* IEGM 1359	*R. opacus* IEGM 249	*R. pyridinivorans* IEGM 1137	*R. rhodochrous* IEGM 107	*R. rhodochrous* IEGM 757	*R. rhodochrous* IEGM 1360	*R. ruber* IEGM 231
Growth substrates	*o-*Xylene, PAHs	*o-*Xylene	Toluene, *o-*xylene	Benzene, *o-*xylene	Naphthalene	-	-	Pyridine	Benzene	Benzene, toluene, PAHs, aniline, pyridine, coumarin
Number of genes coded for										
Dioxygenase (total)	35	30	28	30	69	23	32	41	32	46
Catechol 1,2-dioxygenase	1	1	1	1	4	1	2	3	2	2
Protocatechuate 3,4-dioxygenase	1	1	1	1	1	1	1	1	1	2
Extradiol dioxygenase	2	2	2	2	2	1	2	1	2	6
Catechol 2,3-dioxygenase	0	0	0	0	2	1	0	0	0	1
Gentisate 1,2-dioxygenase	0	0	0	0	3	0	1	2	1	1
Homogentisate 1,2-dioxygenase	1	2	1	2	1	1	2	2	2	1
Benzene 1,2-dioxygenase	0	0	0	0	0	0	0	0	0	1
Anthranilate 1,2-dioxygenase reductase	1	1	1	1	1	0	0	0	0	1
3-Hydroxy anthranilate 3,4-dioxygenase	1	1	1	1	0	0	0	0	0	0
Monooxygenase (total)	66	66	66	66	97	45	63	74	63	72
Phenol monooxygenase	0	0	0	0	0	0	0	0	0	3
Benzoyl-CoA oxygenase/reductase	0	0	0	0	1	0	1	0	1	0
Phenylacetyl-CoA oxygenase/reductase	1	1	1	1	0	0	0	0	0	0
4-Hydroxybenzoate 3-hydroxylase	1	1	1	1	4	2	3	3	3	1
3-Hydroxybenzoate 6-hydroxylase	0	0	0	0	0	0	0	0	0	1
Salicylate 1-hydroxylase	0	0	1	1	0	0	1	0	1	0
Multicopper polyphenol oxidase	1	1	1	1	1	1	1	1	1	4
Dehydrogenase (total)	387	367	350	348	731	256	352	429	348	288
Aldehyde dehydrogenase	13	13	11	12	41	6	15	20	15	5
Succinate-semialdehyde dehydrogenase	1	1	1	1	5	1	1	0	1	1
Hydrolase	125	116	116	116	229	96	115	133	115	123
Phenol hydrolase	0	0	0	0	1	1	0	0	0	0
Amidase	2	2	2	2	2	0	0	0	0	3
Formamidase	1	1	1	1	1	1	1	1	1	0
Fumarylacetoacetate hydrolase family	1	1	0	1	6	0	2	2	2	0
Hydroxymuconic semialdehyde hydrolase	3	2	2	2	0	1	2	2	2	1
Vanillate 3-O demethylase	4	2	2	4	20	4	7	8	7	5
Maleate isomerase	0	0	1	0	2	0	0	0	0	0

**Table 4 molecules-28-02393-t004:** Biodegradation of complex aromatic compounds by *Rhodococcus* strains.

Strain	A Complex Aromatic Compound, Biodegradation Efficiency	Reference
*R. cerastii* IEGM 1278	(RS)-2-(4-(2-Methylpropyl)phenyl)propanoic acid (ibuprofen), 100% degradation of 100 mg/L in the presence of 0.1% (*v*/*v*) *n-*hexadecane within 6 days	[52]
*R. ruber* IEGM 346	[2-(2,6-Dichloroanilino)phenyl]acetic acid (diclofenac), 100% degradation of 0.05 mg/L in the presence of 0.5% (*w*/*w*) D-glucose within 6 days	[53]

**Table 5 molecules-28-02393-t005:** Aromatic compounds used as growth substrates for *Rhodococcus* cells and conditions for their dissolution.

Group of Aromatic Compounds	Compound	Solvent	Heating at Dissolution
Monocyclic aromatic hydrocarbons	Benzene	DMSO	No
	Toluene	DMSO	No
	*o-*Xylene	DMSO	No
PAHs	Naphthalene	Acetone	No
	Anthracene	Acetone	70 °C
	Phenanthrene	Acetone	No
	Benzo[a]anthracene	Acetone	70 °C
	Benzo[a]pyrene	Acetone	70 °C
Aromatic compounds with hydroxy groups	Phenol	Water	No
Aromatic amines	Aniline	DMSO	No
N-heterocyclic aromatic compounds	Pyridine	Water	No
	2-Picoline	Water	No
	3-Picoline	Water	No
	4-Picoline	Water	No
	2,6-Lutidine	Water	No
	2-Hydroxipyridine	Water	No
	4-Hydroxipyridine	Water	No
	Quinoline	70% ethanol	No
Aromatic acids	*o-*Phthalic acid	Acetone	No
	*m-*Phthalic acid	Acetone	No
	*p-*Phthalic acid	Acetone	70 °C
	Salicylic acid	Acetone	70 °C
Aromatic lactones	Coumarin	Acetone	No

## Data Availability

The data presented in this study are available in the Appendix A or can be obtained on request from the corresponding author.

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
