# Peer review of "Rhodococcus Strains from the Specialized Collection of Alkanotrophs for Biodegradation of Aromatic Compounds"

_molecules, 2023, doi:10.3390/molecules28052393_

Round 1
Reviewer 1 Report
In this manuscript, the authors aim to estimate metabolic capabilities of Rhodococcus strains from the IEGM Collection to degrade a variety of aromatic compounds. This manuscript can be accepted for publication after minor revisions to address the following issues:
1. The methods and model conditions adopted to estimate metabolic capabilities of Rhodococcus strains is suggested to more specific.
2. The Figures in the manuscript should be more visual.
3. It is suggested to readjust the selection of keywords.
4. Short sentences will make it easier for readers to understand.
5. It is suggested citation of relevant literature on antibiotic degradation: “A Yb3+/Er3+ co-doped Bi1.95Yb0.04Er0.01V2O8 efficient upconversion glass–ceramic photocatalyst for antibiotic degradation driven by UV–Vis-NIR broad spectrum light”
Author Response
Point-by-point response to the 1 Reviewer’s comments
The manuscript title: Rhodococcus Strains from the Specialized Collection of Alkanotrophs for Biodegradation of Aromatic Compounds
Authors: Anastasiia Krivoruchko, Maria Kuyukina, Tatyana Peshkur, Colin J. Cunningham, and Irina Ivshina
#Reviewer 1 comments
In this manuscript, the authors aim to estimate metabolic capabilities of Rhodococcus strains from the IEGM Collection to degrade a variety of aromatic compounds. This manuscript can be accepted for publication after minor revisions to address the following issues:
- The methods and model conditions adopted to estimate metabolic capabilities of Rhodococcus strains is suggested to more specific.
A: Thank you for the comment. Some corrections about cell concentrations and reagent purities were made in Materials and Methods. Information about controls was corrected in the section 4.4 “Growth experiments”.
- The Figures in the manuscript should be more visual.
A: All Figures were attached as separate files at recommended resolution.
- It is suggested to readjust the selection of keywords.
A: Thank you for the comment. We exclude a keyword “bioremediation” and changed “biodegradation genes” with “functional genes”.
- Short sentences will make it easier for readers to understand.
A: The text was adopted. A number of sentences was shortened. We hope that the text is more readable after these corrections.
- It is suggested citation of relevant literature on antibiotic degradation: “A Yb3+/Er3+ co-doped Bi1.95Yb0.04Er0.01V2O8 efficient upconversion glass–ceramic photocatalyst for antibiotic degradation driven by UV–Vis-NIR broad spectrum light”
A: The suggested reference was added as [10].

Reviewer 2 Report
1. Title. There was already known that Rhodococcus bacteria can degrade aromatic compounds. (Lines 79-81: “It is known that Rhodococcus bacteria can degrade BTEX, PAHs, phenol, 79 phthalates, phthalic esters, pyridine, quinoline, and complex aromatic compounds 80 [6,23,28,29,30–34]. “) Thus, the strong and interesting position of the authors is a study on the big Rhodococcus collection – 133 strains! Please include number of the analyzed strains into the title, namely: “133 Rhodococcus Strains from the Specialized Collection of Alkanotrophs for Biodegradation of Aromatic Compounds”.
2. Abstract. Authors often use word “cells” when they mean “growth”. Please change “cells” for “growth” in the line 16 and delete words “for the growth of Rhodococcus cells” in the line 18. 3. Introduction. Lines 28-78 are some kind of “Introduction into Introduction”. The article is proposed for microbiologist who is interested in the theme highlighted with the key words and title. So, no need to explain origin of the oil & aromatic pollutions and why they are toxic. Please delete the lines 28-78: the article will be much more vivid and specific.4. Results. Results are interesting and helpful for a wide microbiological audience. Please check presentation of results when you mention “cells”. Otherwise it is not enough clear, for example: is “No growth” (figure 3) the same as “No cells” (figure 5)? Is “no cells” equal to “no CFU” or it is some conclusion revealed by microscopy - ?5. Discussion. Discussion is good and interesting.6. Materials and Methods. It would be nice to expand description of the methods (see p. 4 above about presentation of Results).7. Conclusions. At present time “Conclusions” are just an advertisement about the specific professional collection. Please include in the section some information about the presented scientific work. It could include some interesting results which you mentioned in the discussion above (for example, lines 375-377; 427-428; 443-446; 460).
Author Response
Point-by-point response to the 2 Reviewer’s comments
The manuscript title: Rhodococcus Strains from the Specialized Collection of Alkanotrophs for Biodegradation of Aromatic Compounds
Authors: Anastasiia Krivoruchko, Maria Kuyukina, Tatyana Peshkur, Colin J. Cunningham, and Irina Ivshina
#Reviewer 2 comments
- Title. There was already known that Rhodococcus bacteria can degrade aromatic compounds. (Lines 79-81: “It is known that Rhodococcus bacteria can degrade BTEX, PAHs, phenol, phthalates, phthalic esters, pyridine, quinoline, and complex aromatic compounds [6,23,28,29,30–34]. “) Thus, the strong and interesting position of the authors is a study on the big Rhodococcus collection – 133 strains! Please include number of the analyzed strains into the title, namely: “133 Rhodococcus Strains from the Specialized Collection of Alkanotrophs for Biodegradation of Aromatic Compounds”.
A: Thank you for a high estimation of our work and suggestion to be more specific and include the number of studied strains in the title that would highlight a scale of performed experiments. We would like to avoid the indication of the number of strains. About two thousand Rhodococcus isolates are maintained in the Regional Specialized Collection of Alkanotrophic Microorganisms. We are afraid that in the case we specify the number 133, it can result in the conclusion that only these 133 strains can degrade aromatics. This set of strains was more a randomly selected representative sample than a set of strains selected on the base of their known aromatic degrading activities. The aim of the work was to reveal and show the degradation potential of Rhodococcus bacteria towards aromatic pollutants, and the Collection was used as a powerful and available for researchers instrument to search for most promising degraders of these compounds. Other Rhodococcus strains also can degrade aromatic compounds, and the results obtained and presented in this study can help to select biotechnologically important strains, or predict their metabolic properties on the base of species identity and genetics.
- Abstract. Authors often use word “cells” when they mean “growth”. Please change “cells” for “growth” in the line 16 and delete words “for the growth of Rhodococcus cells” in the line 18.
A: Corrected
- Introduction. Lines 28-78 are some kind of “Introduction into Introduction”. The article is proposed for microbiologist who is interested in the theme highlighted with the key words and title. So, no need to explain origin of the oil & aromatic pollutions and why they are toxic. Please delete the lines 28-78: the article will be much more vivid and specific.
A: We completely agree that information about origin, toxicity and a fate of aromatic compounds in the environment are described in literature. The logic and structure of Introduction was ruled by three ideas: (1) to indicate the spectrum and explain the selection of aromatic compounds used as growth substrates in this study, (2) to stress on a problem of chemical contamination, and (3) to introduce the topic of the paper to a wider readership, not only to microbiologists who are interested in microbial degradation of aromatic compounds. The last reason is related to the selected journal Molecules that implies the researchers from other scientific areas than microbiology (chemistry, biotechnology, and those who is searching for novel and efficient methods of elimination of organic pollutants). We have deleted some sentences, tried to shorten Introduction and make the text more readable.
- Results. Results are interesting and helpful for a wide microbiological audience. Please check presentation of results when you mention “cells”. Otherwise it is not enough clear, for example: is “No growth” (figure 3) the same as “No cells” (figure 5)? Is “no cells” equal to “no CFU” or it is some conclusion revealed by microscopy - ?
A: In Figure 5, “no cells” meant abiotic control, the sterile medium without introduction of Rhodococcus cells. It was important to show abiotic control for all compounds since red violet staining could appear not as a result of bacterial oxidation but as a result of abiotic chemical processes. R. ruber IEGM 231 cells meant that Rhodococcus cells were introduced into medium. In Figure 3, “no growth” meant that cells were spread on the surface of agar plates in the presence of organic compounds but they did not grow. Abbreviation CFU was used in experiments with model soil since number of viable PAH-degraders was counted as number of grown colonies on agar plates, and no microscopy was used in this case.
- Discussion. Discussion is good and interesting.
A: Thank you for a high estimation of this part of the work.
- Materials and Methods. It would be nice to expand description of the methods (see p. 4 above about presentation of Results).
A: We have checked Materials and Methods and added some details about cell concentrations and purity of reagents. Additionally, information about controls was corrected in the section 4.4 “Growth experiments”. In growth experiments = determination of spectra of oxidized aromatic compounds by Rhodococcus strains, biotic, abiotic and oligotrophic growth controls were used. In toxicity tests, biotic and abiotic controls were used. In soil experiments, not pre-sterilized soil with introduced Rhodococcus cells and non-sterile soil without introduction of Rhodococcus bacteria were used. Biotic controls were optimal growth medium (LB, LB agar, or mineral medium with glucose) with introduced Rhodococcus cells but without supplementation with aromatic compounds (growth might be detected). Abiotic controls were sterile medium with aromatic substrates but without introduced bacterial cells (growth could not be detected because of the absence of bacterial cells). Controls for oligotrophic growth were mineral medium without aromatic compound supplementation but with introduced Rhodococcus cells. Results of growth experiments = abilities of strains to use aromatic compounds as growth substrates were relevant only if oligotrophic growth was not detected. Rhodococci are oligotrophs and can grow in the presence of trace concentrations of organic compounds, and only ultrapure agar and salts with purity ≥97% guaranteed lack of residual growth of Rhodococcus bacteria on mineral medium without additional growth substrates.
- Conclusions. At present time “Conclusions” are just an advertisement about the specific professional collection. Please include in the section some information about the presented scientific work. It could include some interesting results which you mentioned in the discussion above (for example, lines 375-377; 427-428; 443-446; 460)
A: Conclusions were corrected.
